# COMPACT-A*: SPACE-EFFICIENT FIXED-LENGTH PATH OPTIMIZATION

## ABSTRACT

Several optimization problems seek a path in a state space to minimize a cost function under a length constraint. Traditionally, these are solved by A* search or dynamic programming (DP), as in *Viterbi decoding* for Hidden Markov Models. In all cases, solutions require memory commensurate with a search space that grows linearly in both state space size and path length. In this paper, we propose compact-A*, a framework that limits the growth of the A* priority queue to determine the minimum cost for a path of predetermined length in a space-efficient manner and then constructs such a path by a divide-and-conquer strategy that eliminates the memory overhead. We apply compact-A* to Viterbi decoding and further highlight its generality with an application to V-optimal histogram construction. Our experimental results demonstrate significant improvements over state-of-the-art solutions in runtime and memory consumption.

## 1 INTRODUCTION

Several problems call for finding a sequence of given length $L$ over a state space of size $n$ that optimizes a cost function; they are conventionally solved by A* search or dynamic programming (DP). Prominent examples are *Viterbi decoding*, i.e., finding a most likely sequence of hidden states—a *Viterbi path*—in a Hidden Markov Model (HMM) to explain an observed event sequence of length $L$ (Viterbi, 1967; 2006); finding a sequence of $L-1$ boundaries—a *V-optimal histogram* ('V' for variance)—that partitions a numeric sequence minimizing the total squared error of representing each partition by a constant value (Jagadish et al., 1998); and identifying the most probable syntactic structure—a Viterbi parse tree—that generates a length-$L$ input sequence of tokens or words under a probabilistic context-free grammar (PCFG) (Klein & Manning, 2003; Huang et al., 2012).

Such algorithms iteratively derive solutions to a subproblem of length $k+1$ from solutions to subproblems of length $k$ in $\mathcal{O}(n^2L)$ time. A* prioritizes subproblems in a *best-first* manner using a priority queue with an $\mathcal{O}(nL \log nL)$ overhead, while DP prioritizes them in a *breadth-first* manner by length. In terms of space, we may address the problem as a *shortest path* problem over a graph of $n$ vertices and $m$ edges replicated over $L$ layers, one for each step, *memoize* partial solutions, and backtrack over steps to build the optimal path after finding the optimal cost. Still, this strategy uses $\mathcal{O}(nL + m)$ space. We may instead work *in-place* in $\mathcal{O}(n + m)$ space, without replication, discarding processed subproblem solutions. Still, in that case we need to recursively rerun the algorithm from scratch $L-1$ times to build the optimal path after finding the optimal cost, incurring a $\mathcal{O}(n^2L^2)$ *time complexity* overhead instead. A recently proposed reformulation of the DP solution (Ciaperoni et al., 2022; 2024) constrains space to $\mathcal{O}(n + m)$ and runs in $\mathcal{O}(n^2L \log L)$ time, but evaluates subproblems exhaustively by breadth-first search, harming scalability. In practicality, only a few subproblems aid the solution. A*-based algorithms exploit this fact, yet need to *(i)* maintain a priority queue and *(ii)* memoize optimal choices per step to enable backtracking. These needs take only $\mathcal{O}(n)$ space for the shortest-path problem, in which path length is *arbitrary* (Russell & Norvig, 2010), yet grow to $\mathcal{O}(nL)$ space when path length $L$ is *predetermined* (Klein & Manning, 2003; Huang et al., 2012).

In this paper, we introduce compact-A*, a framework for space- and time-efficient optimization of *fixed*-length sequences. Compact-A* controls the priority queue size by novel search strategies we propose, and avoids memoization through divide-and-conquer, yielding $\mathcal{O}(n)$ space use. We apply compact-A* to Viterbi decoding and V-optimal segmentation and test it on real and synthetic data, showing gains in memory use and runtime vs. prior work, especially under skewed cost distributions.

## 2 BACKGROUND AND RELATED WORK

**Dynamic programming** (DP) (Bellman, 1966) explores the problem space exploiting an *optimal substructure* property, by which a globally optimal solution can be assembled from locally optimal solutions, to solve a problem by recursively *expanding* partial solutions in a breadth-first manner. **Best-first-search** algorithms (Pearl, 1984) may also exploit optimal substructures, as dynamic programming does (Sniedovich, 2006), to find a minimum-cost path from a start to an end node by repetitively *expanding*, i.e., visiting the neighbors of, the *most promising* unexpanded visited node. The A\* algorithm (Russell & Norvig, 2010; Foead et al., 2021), an instance of best-first-search, prioritizes paths by a path cost estimation heuristic that is *admissible* (i.e., never overestimates the cost of a path) and *consistent* (i.e., never estimates the cost of a path as greater than the cost via an intermediary node) (Russell & Norvig, 2010); the search may proceed from both start and end by bidirectional search (Pohl, 1969). A\* generalizes Dijkstra's algorithm (1959), by which the heuristic cost estimate of any unexplored path is 0. We stress that finding a minimum-cost path of *fixed length* requires tracking both length and cost, while for *arbitrary length*, tracking cost suffices.

**Hidden Markov Models** (HMMs) explain observation sequences. An HMM comprises a set of $K$ hidden states, each with probabilities to be an initial state, transition to other states, and emit an observation. *Decoding* seeks a sequence of states most likely to generate a sequence of observations:

**Problem 1** (Decoding). *Given an HMM and a sequence of $T$ observations $Y = \{y_1, y_2, \ldots, y_T\}$, find the sequence of hidden states $Q = \{s_1^*, s_2^*, \ldots, s_T^*\}$ that maximizes the likelihood $P(Q, Y)$.*

The Viterbi algorithm (1967; 2006) solves Problem 1 optimally by DP; it finds application from networking and telecommunications (Viterbi, 2006) to speech recognition (Gales & Young, 2007; Braun et al., 2020), where it serves to find the most probable transcription for an input acoustic signal, or for forced alignment, the task of aligning orthographic transcriptions to audio recordings. In modern speech-recognition systems, the Viterbi algorithm runs on the composition of several small HMMs in which states represent words and their phonemes, to find the best transcription of a spoken utterance. However, this algorithm raises high memory and runtime requirements. A recent work (Jo et al., 2019) on HMM-based isolated word recognition employed a search heuristic, without proving its correctness. Another recent work (Ciaperoni et al., 2022) enhances the space efficiency of decoding at the cost of a runtime overhead. Other works reduce the state space representation for particular classes of HMMs (Siddiqi & Moore, 2005; Felzenszwalb et al., 2003). Still, time complexity remains high for problem instances with large state space and long observation sequences, while improving on it is a difficult undertaking, as indicated by derived lower bounds (Backurs & Tzamos, 2017). Klein & Manning (2003) and Huang et al. (2012) apply A\*-like policies to enhance Viterbi's time efficiency in the context of PCFG parsing, yet neglect space efficiency. Compact-A\* offers a formulation of Viterbi decoding that is both time-efficient and parsimonious.

**Histogram construction** calls to segment a data series to a predetermined number of buckets, each with one representative, to minimize the overall representation error:

**Problem 2** (Histogram Construction). *Given $I = \{x_1, \ldots x_n\}$, $x_i \in \mathbb{R}$, and $B \in \mathbb{Z}^+$, find a segmentation (or histogram) $H_B$ of $I$ into $B$ non-overlapping subsequences (or buckets) $I_b$ with associated bucket representatives $\hat{x}_b$, $b \in \{1, \ldots, B\}$, that minimizes error function $E_I(H_B)$.*

Problem 2 is central in data summarization (Halim et al., 2009). We focus on *V-optimal* histogram construction (Jagadish et al., 1998), i.e., Problem 2 with $E_I(H_B) = \sum_{b=1}^{B} E_b$, where $E_b = \sum_{x_i \in I_b}(x_i - \hat{x}_b)^2$, and $\hat{x}_b$ is the mean of values in bucket $I_b$. This extensively studied problem (Guha et al., 2006) is solved optimally by a DP algorithm (Jagadish et al., 1998) with quadratic dependence on $n$. Compact-A\* offers significant gains in histogram construction and is extensible to any monotonic and distributive error measure $E_I(H_B)$ (Karras & Mamoulis, 2008).

**Semirings and Dioids** (Gondran & Minoux, 2008) A *semiring* is a 5-tuple $(D, \oplus, \otimes, , )$, where $D$ is a non-empty set, $\oplus$ is a binary, associative, and commutative operator, $\otimes$ is a binary and associative operator, is a neutral element for $\oplus$ (i.e., $x \oplus = x$, for all $x \in D$), is a neutral element for $\otimes$ (i.e., $x \otimes = \otimes x = x$, for all $x \in D$), the operator $\otimes$ distributes over $\oplus$ and is absorbing for $\otimes$ (i.e., $x \otimes = \otimes x = $, for all $x \in D$). A *selective dioid* is a semiring in which $\oplus$ is also selective (i.e., $(x \oplus y = x) \vee (x \oplus y = y)$, for all $x, y \in D$). Selective dioids provide an abstract expressive framework for shortest-path and DP problems (Mohri, 2002; Huang, 2008; Tziavelis et al., 2020).

## 3 THE COMPACT-A* FRAMEWORK

We define compact-A* and apply it to Viterbi decoding (§ 3.1) and histogram construction (§ 3.2). Compact-A* finds a given-length sequence that optimizes a cost measure. Contrariwise to dynamic programming, which visits subproblems in a fixed order even if several of them do not contribute to the final solution, compact-A* solves subproblems in a best-first fashion. Figure 1 shows an example.

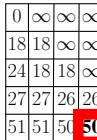 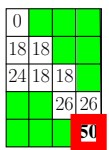

Figure 1: Each cell records the cost of a sub-problem; classic DP solves all sub-problems; compact-A* finds the same solution, but avoids considering sub-problems in green.

Compact-A* starts out with the following components:

1. A *data space* $\mathcal{X}$ of $n$ elements endowed with a concept of *eligible sequence* $(x_j)$, $x_j \in \mathcal{X}$, $j \in \{1, \ldots, \ell\}$, where $\ell$ is the length of $(x_j)$; a sequence is eligible if $i - 1 \in \mathcal{N}(i)$ for all $i \in \{2, \ldots, \ell\}$ and a given *neighborhood* function $\mathcal{N}(\cdot)$.
2. The *set* $\mathbb{X}$ of all eligible sequences in $\mathcal{X}$.
3. A *gap function* $G(j, i)$ associating a value with the transition from item $j$ to item $i$.
4. A selective dioid $\mathcal{D} = (D, \oplus, \otimes, , )$, which is used to express the *value function* for a sequence: $f(\mathbf{x}) = \bigotimes_{j=1}^{j=L-1} G(j, j+1)$.
5. A *problem* that seeks an eligible sequence $\mathbf{x}^* \in \mathbb{X}$ of *length* $L$ and *optimal* value $f(\mathbf{x}^*)$; sequences are compared via the $\oplus$ operator.
6. A recursive *function* $\mathsf{Opt}(i, \ell)$ that stores the *optimal* value for an eligible sequence of length $\ell$ ending at item $x_i \in \mathcal{X}$.
7. A *solution* to the problem in Item 5 by DP over sequences of increasing length from $\mathcal{X}$.

The solution in Item 7 finds an eligible sequence of $L$ data items $\mathbf{x}^* = \{x_1^*, x_2^*, \ldots, x_L^*\}$ that optimizes $\mathsf{Opt}(\cdot, L)$; the selective dioid properties, in particular distributivity, guarantee correctness. The DP computation takes the form:

$$\mathsf{Opt}(i, \ell) = \bigoplus_{j \in \mathcal{N}(i)} \{\mathsf{Opt}(j, \ell - 1) \otimes G(j, i)\}. \tag{1}$$

The recursion of Equation (1) requires $\Theta(n^2 L)$ time and $\Theta(nL)$ space, iterating over items $i$ and lengths $\ell$ and storing, for each $(i, \ell)$ pair, a predecessor needed to backtrack the optimal sequence. The entailed solution may be implemented by either DP or *token passing* (Young et al., 1989); both calculate *all* solutions of length $\ell$, $\mathsf{Opt}(\cdot, \ell)$, before those of length $\ell+1$, $\mathsf{Opt}(\cdot, \ell+1)$, by *breadth-first search*. While DP draws from solutions of length $\ell$ to build each solution of length $\ell + 1$, token passing broadcasts a *token* for each solution of length $\ell$ to its continuations of length $\ell + 1$. In both cases, solutions solidify at length $\ell$ before moving to length $\ell + 1$.

Compact-A* abolishes this *breadth-first* orientation in favor of a *best-first* one; it organizes sub-problem solutions (represented by tokens) in a priority queue $\mathbf{Q}$ in which it initially inserts all tokens $(\cdot, 1)$. Thereafter, in each step, it selects the most promising token $(j, \ell)$ from $\mathbf{Q}$ and, for each eligible successor $(i, \ell + 1)$ not already extracted from $\mathbf{Q}$, it computes $\mathsf{Opt}(j, \ell) + G(j, i)$ and updates the priority of $(i, \ell + 1)$ in $\mathbf{Q}$ accordingly. compact-A* resembles Dijkstra's shortest-path algorithm (1959); however, whereas Dijkstra minimizes a cost objective regardless of length (i.e., number of steps), compact-A* optimizes the objective under a *fixed*-length constraint. In the worst case, it examines all sequence continuations for each length, hence takes $\mathcal{O}(nL(n + \log nL))$ time, where the $\log nL$ term expresses the overhead of maintaining the priority queue, more compactly $\mathcal{O}(nL(n + \log L))$. However, in practice, it gains performance as it quickly derives tokens corresponding to DP table cells without considering all possible paths and does not produce some tokens at all. As we will see in Section 4, this pruning capacity results in significant savings, particularly in real problem instances. In problem-tailored compact-A* variants, we anticipate the cost a sequence may obtain as it expands and prioritize tokens accordingly. We also derive bidirectional-search variants that produce both prefixes and suffixes of sequences until they reach the target length.

In the best case, compact-A* may produce only $L$ tokens. Nevertheless, the priority queue $\mathbf{Q}$ may reach size $\Theta(nL)$, holding one token for each state-length pair. Additionally, we need to memoize the subproblem solution each token represents even after we pop it from $\mathbf{Q}$, to enable *backtracking* the solution sequence. To guarantee $\mathcal{O}(n)$ space complexity, we employ the following measures.

**Controlling the priority queue size.** First, to keep the size of $\mathbf{Q}$ in $\mathcal{O}(n)$, when the best-first-search queue $\mathbf{Q}$ exceeds a predefined size threshold $\theta$ (or memory budget) upon inserting a token $(s_j, t)$, we invoke a *containment mechanism*, which is either a tailored *depth-first-search* (DFS) mechanism, or a mechanism that we propose, *earliest-first search* (EFS).

**DFS.** By the DFS mechanism, we pick the lowest-cost token $(s_j, t^*)$ from the set of tokens for $s_j$, $\mathcal{S}^j = \{(s_j, \cdot)\}$, and generate all its derivative tokens via a DFS traversal of the HMM graph $G$ starting at $s_j$. Each DFS branch terminates upon reaching the last frame or upon injecting into the token set of a state $s_{j'}$ a token that either displaces or yields to a pre-existing one, without increasing memory usage in either case. This DFS mechanism identifies and propagates middle pairs along explored paths as usual and ensures space complexity $\mathcal{O}(n)$ by constraining the size of $\mathbf{Q}$ within $\theta$.

**EFS.** Small $\theta$ values in the DFS mechanism may cause excessive DFS calls, reprocessing the same tokens multiple times and slowing runtime, even compared to that of regular dynamic programming which processes all $nL$ tokens. To address this predicament, we propose an alternative search mechanism that processes each token at most once, *Earliest-First Search* (EFS). EFS processes tokens in increasing time frame order, starting form the earliest frame represented in the priority queue, until the gap between the earliest and latest time frames in the queue falls below a user-specified constant $\Delta_{\mathbf{Q}}$. In effect, the queue size is bounded by $\Delta_{\mathbf{Q}} n$ and best-first search resumes. While the EFS mechanism generates and must store additional tokens, at most $2n - 1$ additional tokens will co-exist in the queue, occupying the earliest time frame and its successor. In effect, EFS ensures an overall queue size of $\mathcal{O}(n)$.

**Reconstructing the optimal path.** Second, instead of memoizing subproblem solutions for backtracking the final solution path, we construct that path by a *divide-and-conquer* strategy in $\mathcal{O}(n)$ space and $\mathcal{O}(n^2 L \log L)$ time, as in (Binder et al., 1997; Ciaperoni et al., 2022; 2024). Upon reaching the *middle* frame of a solution path, we record, with each subsequent token, the edge at that middle frame (or *middle pair*). After establishing the best solution at the last frame, we recursively rerun the algorithm on the $L/2$-hop predecessors and successors of that token's middle pair to construct the entire sequence. We retrieve middle pairs in orderly fashion, as in an in-order tree traversal (Ciaperoni et al., 2022) in $\mathcal{O}(n^2 L \log L)$ time and $\mathcal{O}(n)$ space, given that the size of $\mathbf{Q}$ is also bounded by $\mathcal{O}(n)$.

## 3.1 THE MINT ALGORITHM

The Viterbi algorithm selects a sequence of $T$ states $Q = \{s_1^*, s_2^*, \ldots, s_T^*\}$ from a universe of $K$ HMM states $S = \{s_1, s_2, \ldots, s_K\}$ that is *most likely* to have generated a sequence of $T$ observations $Y = \{y_1, y_2, \ldots, y_T\}$. $Q$ is called *Viterbi path*. By the Markov property, the likelihood to be in a state depends only on the previous state. Therefore, the Viterbi algorithm uses the DP recursion:

$$\mathbf{T}[s_i, 1] = \pi_{s_i} \cdot B_{s_i, y_1},$$
$$\mathbf{T}[s_i, t] = \max_{s_h \in \mathcal{N}_{in}(s_i)} \{\mathbf{T}[s_h, t-1] \cdot A_{s_h, s_i}\} \cdot B_{s_i, y_t} \tag{2}$$

where $\mathbf{T}[s_i, t]$ stores the probability of the most likely path ending at state $s_i$ in $t$ steps, or *time frames*, $\mathcal{N}_{in}(s_i)$ is the set of in-neighbors of $s_i$, $\pi_i$ is the initial probability of $s_i$, $A_{s_h, s_i}$ is the probability of transiting from state $s_h$ to state $s_i$ on a directed graph $G$ capturing eligible transitions in the HMM, and $B_{s_i, y_t}$ is the probability of observing $y_t$ at state $s_i$. This setting suits compact-A* as follows:

1. The *data space* $\mathcal{X}$ is the universe of $K$ hidden states $S = \{s_1, s_2, \ldots, s_K\}$ and an *eligible sequence* $(x_j)$, $x_j \in \mathcal{X}$, $j \in \{1, \ldots, \ell\}$ is a path of consecutive states in $G$.

2. The *set* $\mathbb{X}$ of all eligible sequences in $\mathcal{X}$ is the set of all possible paths in the given HMM.

3. The *gap function* $G(j, i)$ is $A_{s_j, s_i} B_{s_i y_i}$, or, in log-probabilities, as $\log A_{s_j, s_i} + \log B_{s_i y_i}$.

4. The selective dioid is $([0, 1], \max, \cdot, 0, 1)$, or, in the domain of log-probabilities, $([-\infty, 0], \max, +, -\infty, 0)$. Thus, the *value function* $f$ assigns probabilities to paths, given the sequence of observations $Y = \{y_1, y_2, \ldots, y_T\}$; the probability that $Y$ is generated by a

sequence of hidden states $Q = \{s_1, s_2, \ldots, s_T\}$ is $P(Q, Y) = \pi_{s_1} \cdot B_{s_1 y_1} \prod_{i=2}^{T} A_{s_{i-1} s_i} \cdot B_{s_i y_i}$, where $\pi(s_1)$, $A_{s_{i-1} s_i}$, and $B_{s_i y_i}$ are defined as above.

5. The *problem* seeks an eligible sequence of states $Q$ of length $T$ that best explains the given sequence of observations $Y$, i.e., maximizes probability ($\oplus = \max$).

6. The recursive *function* $\mathrm{Opt}(i, \ell)$ that stores the *optimal* value for an eligible sequence of length $\ell$ that ends at data item $x_i \in \mathcal{X}$ is the function $\mathbf{T}[s_i, t = \ell]$.

7. The *solution* by DP over sequences of increasing length from $\mathcal{X}$ is given by Equation (2).

The recursion of Equation (2) requires $\mathcal{O}(K^2 T)$ time and $\mathcal{O}(KT)$ space, as it iterates over states $s_i$ and time frames $t$. For the sake of efficiency and accuracy, we replace products of likelihoods by sums of log-likelihoods. In case the structure of $G$ is known, we iterate only over states $s_h$ that link to state $s_i$, hence visit each HMM graph edge only once; then time complexity becomes $\mathcal{O}((K + |E|)T)$.

The Viterbi algorithm and its *token passing* variant (Young et al., 1989) operate by *breadth-first search*. MINT (Time Efficient Viterbi) replaces this strategy with *best-first* search. It organizes partial solutions in a priority queue $\mathbf{Q}$ and expands the most promising one in each step, as in (Klein & Manning, 2003). Like Viterbi, it derives $\mathbf{T}[s_i, t + 1]$ entries from $\mathbf{T}[s_h, t]$ entries. Still, while Viterbi calculates *all* $\mathbf{T}[\cdot, t]$ entries before $\mathbf{T}[\cdot, t + 1]$ entries, MINT first inserts all tokens $(s_h, 1)$ in $\mathbf{Q}$ and then iteratively pops the most promising token $(s_h, t)$ from $\mathbf{Q}$ and, for each *outgoing* neighbor $s_i \in \mathcal{N}_{out}(s_h)$ such that $(s_i, t + 1)$ has not been extracted from $\mathbf{Q}$, it computes $\mathbf{T}[s_h, t] \cdot A_{s_h, s_i} \cdot B_{s_i, y_{t+1}}$ as a provisional $\mathbf{T}[s_i, t + 1]$ value and updates the priority of $(s_i, t + 1)$ accordingly. Thanks to the *monotonicity* of $\mathbf{T}[\cdot, \cdot]$ along a path, $\mathbf{T}[s_i, t]$ satisfies Equation (2) when $(s_i, t)$ is popped from $\mathbf{Q}$. In the worst case, MINT visits each HMM edge once per time frame, hence takes $\mathcal{O}((K \log KT + |E|)T)$ time, where $\log KT$ expresses the priority queue overhead (Fredman & Tarjan, 1987). In practicality, it never considers some HMM edges. In the following, we illustrate four MINT variants.

**Standard MINT.** In more detail, to achieve the $\max_{s_i} \mathbf{T}[s_i, T]$ objective, we *minimize* the positive path log-likelihood, $-\log P(Q, Y) \geq 0$, defined as *cost* of a path. For a given path $Q$ ending at state $i$ at frame $t$, we define its priority as $p(s_i, t) = -\log P(Q, Y)$, with $Q = \{s^1, s^2, \ldots, s^t\}$, such that $s^t = s_i$ and $Y = \{y_1, y_2, \ldots, y_t\}$. Algorithm 1 in Appendix D gives the pseudocode of the plain MINT variant. First we insert in $\mathbf{Q}$ a token for each state in the first frame with priority $-\pi_{s_i} - \log B_{s_i, y_1}$. If a start state $s$ is given, we only enqueue a token for $s$ at $t = 0$. The main loop iterates until reaching the last frame. In each iteration, we dequeue from $\mathbf{Q}$ the top token $(s_h, t)$, add it to a set $\mathbf{V}$ of visited tokens, compute the cost of reaching $(s_i, t + 1)$ via $(s_h, t)$ for each out-neighbor $s_i$ of $s_h$ that has no such token in $\mathbf{V}$, and insert or update $(s_i, t + 1)$ in $\mathbf{Q}$ to the lowest known cost. When a pair $(s, T)$ is dequeued, no path spanning $T$ frames at lower cost exists, hence we return its cost as the Viterbi path log-likelihood $\max_{s_i} \mathbf{T}[s_i, T]$. All proofs are in Appendix A.

**Proposition 1.** *Standard MINT is correct.*

To return the optimal path, MINT by defaults appends a path to each token and, when updating the priority of $(s_j, t + 1)$, also updates the corresponding path to $s_j$. An alternative implementation, MINT-Backtracking, stores only the *predecessor* of each token, retains all explored tokens, and at the end constructs the optimal path by *backtracking*. Next, we present three extensions to MINT.

**MINT Bound.** We propose a variant of MINT that orders tokens by lower-bounding the path cost from each token $(s_i, t)$ until the final frame $T$ using a *lower bound* $\hat{c}_1$ on the cost for moving from one frame to another for the remaining $T - t$ frames. We call the ensuing algorithm MINT Bound.

We first insert all states (or the source state, if given) in $\mathbf{Q}$ with $p(s, 1) = T \cdot \hat{c}_1$. As the search proceeds, we replace lower bounds with exact costs. The priority of $(s_i, t)$ is $p(s_i, t) = -\log P(Q, Y) + (T - t) \cdot \hat{c}_1$, $Q$ being the current optimal path ending at $s_i$ in $t$ frames; we compute the priority of a neighbour $s_j$ for insertion or update in $\mathbf{Q}$ as $p(s_i, t) - \hat{c}_1 - \log A_{s_i, s_j} - \log B_{s_j, y_{t+1}}$. Upon reaching the last frame, all lower bounds capture exact costs. Setting $\hat{c}_1$ as the lowest value of $-\log A_{s_i, s_j} - \log B_{s_j, y_{t+1}}$ over all edges $(s_i, s_j)$ at any $t$, found by pre-processing, ensures correctness.

**Proposition 2.** *MINT Bound is correct.*

MINT Bound encases information from unexamined time frames in the BestFS criterion and further explores already well-explored paths without compromising correctness.

**Bidirectional MINT** obtains more efficiency by bidirectional search (Pohl, 1969), explore solutions both forward from the start and backward from the end time frame. We denote the graph in which

all edges are reversed in direction as $G_{rev}$. In each direction, the search proceeds as in Standard MINT, yet with two priority queues, $\mathbf{Q}_f$ for forward search and $\mathbf{Q}_b$ for backward search. If an initial state is given and a final state is not given, we find, in pre-processing, all states $\mathcal{S}(T)$ reachable from the initial state in $T$ frames, and initiate $\mathbf{Q}_b$ with those. Algorithm 2 in Appendix D shows the pseudocode, which assumes that both an initial and final state are given. In each iteration, we expand both[1] searches, handling queues as in Standard MINT. Upon extracting $(s_i^f, t^f)$ from $\mathbf{Q}_f$ and $(s_i^b, t^b)$ from $\mathbf{Q}_b$, we update the associated visited-token sets, $\mathbf{V}_f$ and $\mathbf{V}_b$, and store associated costs in arrays $d_f$ and $d_b$. We then consider tokens $(s_j^f, t^f + 1)$ for all neighbors $s_j^f$ of $s_i^f$ in $G$ and tokens $(s_j^b, t^b - 1)$ for all neighbors $s_j^b$ of $s_i^b$ in the graph $G_{rev}$. In Lines 10 and 15 we omit the details, which are found in Algorithm 1, Lines 30–39. When the two sides meet generating a path of length $T$, we update the hitherto best path cost $\mu$, if the newly found path improves on it. Such a path is provably optimal, hence the algorithm terminates, when the sum of costs of tokens dequeued from $\mathbf{Q}_f$ and $\mathbf{Q}_f$ exceeds $\mu$. To avoid double-counting emission probabilities, we add the emission probability of the last vertex visited while building a path only in the priority of $\mathbf{Q}_f$, thus the cost of any path through $(w, t)$ is $d_f[(w, t)] + d_b[(w, t)]$.

**Proposition 3.** *Bidirectional MINT is correct.*

**Bidirectional MINT Bound** combines Bidirectional MINT and MINT Bound in one algorithm that searches in both directions and lower-bounds the total $T$-frames-long path costs used as priority values in both queues. The search in both directions follows the order determined by the cost of arriving to a state in a given number of steps (frames) plus a lower bound on the cost of arriving from there to the the end of the path. The single-frame lower bound for the forward search is as in MINT Bound. For backward search, it is the lowest cost of moving from a frame to the next one in the reverse HMM graph $G_{rev}$, i.e., the lowest value of $-\log A_{s_j, s_i} - \log B_{s_j, y_t}$ over all edges $(s_j, s_i)$ in $G_{rev}$. We obtain both bounds by pre-processing with a single graph traversal. The correctness of Bidirectional MINT Bound follows from the correctness of MINT Bound and Bidirectional MINT.

**Linear-space MINT (MINT-LS)** Here, we propose a compact-A* space-efficient variant, MINT-LS, which guarantees $\mathcal{O}(K)$ space by keeping the size of $\mathbf{Q}$ in $\mathcal{O}(K)$ by a *containment* strategy and reconstructs the solution by a *divide-and-conquer* strategy. Algorithm 5 in Appendix D presents MINT-LS, which invokes the DFS subroutine of Algorithm 4, while storing, with each token $(s_i, t)$, its predecessor state and the running middle pair for its path. These details help identify the subproblems to be solved recursively. Upon reaching the final frame (and the final state, if specified), it extracts the middle pair associated with the solution path and reruns recursively among $N_p$-hop predecessors of the middle pair in as many preceding frames and among $N_s$-hop successors in as many following frames, identified through breadth-first search (Lines 21 and 27). MINT-LS avoids storing tokens when the queue size reaches a threshold $\theta$. Instead, it invokes the DFS subroutine, which only stores tokens either as replacements of others or at the last frame. Algorithm 6 in Appendix D presents MINT-LS+, which applies our EFS strategy using an auxiliary queue $\mathbf{Q}_t$ to prioritize tokens by time frame; it keeps track of the earliest and latest time frame in $\mathbf{Q}_t$ and toggles between BestFS and EFS as required (Lines 41–47). As results in Section 4.1 and Appendix C show, these strategies ensure low space consumption. When $\theta$ is large enough, the DFS strategy is preferable; otherwise, EFS is preferred. MINT-LS naturally combines with bidirectional search.

### 3.2 THE TECH ALGORITHM

We apply compact-A* to V-Optimal histogram construction (Jagadish et al., 1998), outlined in Section 2. A V-Optimal histogram uses the mean value $\hat{x} = \frac{1}{i-j+1} \sum_{k=j}^{i} x_k$ as a representative to minimize the *Euclidean* error in a bucket $I_b$ extending from the $j^{\text{th}}$ to the $i^{\text{th}}$ value in the sequence, $E(j, i) = \sum_{k=j}^{i} (x_k - \hat{x})^2$. The total error is aggregated over all buckets. V-Optimal algorithms use incremental sums and sums of squares, $S$ and $SS$, respectively, to obtain any $E(i, j)$ as:

$$E(j, i) = (SS[i] - SS[j-1]) - \frac{(S[i] - S[j-1])^2}{(i-j+1)}. \tag{3}$$

The algorithm finds, for each combination $(i, b)$ of a value index and number of buckets, the cost of the optimal $b$-bucket histogram covering the first $i$ values in the sequence, as:

---

[1]A more refined strategy would choose which side to expand, representing an opportunity for future work.

$$E^*(i,b) = \min_{1 \le j < i} E^*(j, b-1) + E(j+1, i), \qquad (4)$$

$E^*(i,b) = 0$ for $i \le b$ and $E^*(i,1) = E(1,i)$. After finding the optimal cost $E^*(n, B)$ for a length-$n$ sequence and $B$ buckets, we backtrack the histogram. The problem maps to compact-A* as follows:

1. The *data space* $\mathcal{X}$ is the data series $I = \{x_1, \ldots x_n\}$ and an *eligible sequence* $(x_j)$, $x_j \in \mathcal{X}$, $j \in \{1, \ldots, \ell\}$ comprises *ordered* segment boundaries $(x_{i_1}, x_{i_2}, \ldots, x_{i_k})$.
2. The *set* $\mathbb{X}$ of eligible sequences in $\mathcal{X}$ includes all ordered sequences ending at $x_n$.
3. The *gap function* $G(j, i)$ is the bucket error $E(j, i)$.
4. The selective dioid is $(\mathbb{R} \cup \{+\infty\}, \min, +, +\infty, 0)$. Thus, the *value function* $f : \mathbb{X} \to \mathbb{R}$ assigns an approximation error $\sum_{b=1}^{B} E_b$ to a histogram.
5. The *problem* seeks an error-minimizing eligible histogram of $B$ boundaries ($\oplus = \min$).
6. The recursive *function* $\mathsf{Opt}(i, \ell)$ that stores the *optimal* value for an eligible sequence of length $\ell$ ending at item $x_i \in \mathcal{X}$ is the function $E^*(i, b = \ell)$.
7. The *solution* by DP over sequences of increasing length from $\mathcal{X}$ is given by Equation (4).

V-OPT histogram construction by DP (Jagadish et al., 1998) requires $O(n^2 B)$ time and $O(nB)$ space. We discuss compact-A* variants, TECH (Time-Efficient Histogram), aligned to the variants of MINT.

**Standard TECH.** Algorithm 3 presents Standard TECH, which, like MINT, employs a priority queue $\mathbf{Q}$ to prune computations, where the priority of entry $(i, b)$ is the cost of the $b$-bucket histogram for the first $i$ values, $p(i, b) = E^*(i, b)$. After computing the $S$ and $SS$ arrays, used to compute the error measure by Equation (3), in each iteration, TECH dequeues the pair $(i, b)$ of lowest error, adds it to a set $\mathbf{V}$ of visited tokens, and, provided $b < B$, computes via $(i, b)$ the error for each pair $(j, b+1)$ with $j > i$ that is not in $\mathbf{V}$ and inserts or updates $(j, b+1)$ in $\mathbf{Q}$ accordingly; thereby, it explores possible next buckets. We do not iterate over $(j, b+1)$ pairs for all $j > i$, but stop at $j = n - B + b + 1$ since there must be at least $B - b - 1$ values after $j$ to make $B$ buckets in total. The algorithm terminates after it dequeues $(n, B)$. Correctness follows as in Proposition 1: after $(n, B)$ is dequeued, there can be no lower-cost histogram of the same series and $B$. For further pruning, we use an upper bound $UB[i, b]$ on the cost of a $(B - b)$-bucket histogram for the series $\{i + 1, \ldots n\}$, derived in Proposition 4 below. If, after visiting $(i, b)$, we find $j^* > i$ such that $E(i, j^*) \ge UB[i, b]$, we eschew computing $E(i, j)$ for $j \ge j^*$, as we have already exceeded the upper bound on the error therefrom.

**TECH Bound.** This variant uses bounds on the cost of a $B$-bucket histogram instead of the cost of a partial histogram with $b \le B$ buckets. Given $(i, b)$, we partition the series $\{i + 1, \ldots n\}$ in $B - b$ equal-width buckets. The minimum error among such buckets is a lower bound to the V-optimal histogram cost, while the sum of those errors is an upper bound, which we may use for pruning.

**Proposition 4.** *The minimum error* $\min_b E_b$ *among* $b$ *buckets of an equal-width partitioning a sequence* $I$ *is a lower bound to the V-optimal histogram cost and* $\sum_b E_b$ *is an upper bound.*

TECH Bound adds the lower bound $LB[i, b]$ to the priority of each pair $(i, b)$. Upon arrival at the end of the series, it outputs the same cost as Standard TECH. We find these bounds by building equal-width histograms in a pre-processing step. Correctness follows as in the case of MINT Bound.

Other TECH variants work by analogy to MINT variants. **Bidirectional TECH** applies forward and backward search. When we pop a pair $(i, b)$ from $\mathbf{Q}_f$, we consider entries $(j, b+1)$ with $j > i$, and, if the backward search has already visited $(j, B - b - 1)$, we check whether the cost $d_f[(i, b)] + E(i, j) + d_b[(j, B - b - 1)]$ improves upon the current best cost $\mu$, and likewise in backward search. The search terminates when the sum of the costs of pairs from both queues exceeds the current best cost $\mu$. **Bidirectional TECH Bound** combines Bidirectional TECH with TECH Bound, with reversed lower bounds to consider sequences of the form $\{1, \ldots i\}$ rather than $\{i + 1, \ldots n\}$. **TECH-LS** variants limit space needs; in place of a *middle pair* of states, they detect a *middle bucket* that splits the data series in halves. Guha (2005) applied such a space-saving solution on DP.

## 4 EXPERIMENTS

We experimented[2] on a $2 \times 12$ core Xeon E5 2680 v3 2.50 GHz machine with 128 GB RAM. For Viterbi decoding, we use as baselines the edge-aware Viterbi algorithm (§3.1), a recomputation-based

---

[2]Implementation and data available at https://anonymous.4open.science/r/BestFirst.

space-efficient variant (Ciaperoni et al., 2022), checkpoint Viterbi (Tarnas & Hughey, 1998), which segments the sequence into $\sqrt{T}$ parts, and SIEVE variants (Ciaperoni et al., 2022), including SIEVE-Middlepath, Standard SIEVE (partitioning the state space recursively), and SIEVE-Hyperloglog (using approximate predecessor/successor counts (Flajolet et al., 2007). For histogram construction, the baseline is the DP algorithm of Jagadish et al. (1998) (§ 3.2). Appendix B details data, measures, and parameter choices. Appendix C presents further experiments.

## 4.1 RESULTS ON VITERBI DECODING

**Real data, runtime vs. $T$ and $K$.** Figure 2 plots results on forced-alignment and standard decoding with real data. Here, MINT achieves savings of up to three orders of magnitude over Viterbi by focusing on promising paths. Enlarging the state space via larger snowball samples of the HMM does not affect MINT, but heavily impacts Viterbi. MINT-LS variants also gain up to three orders of magnitude lower runtime than Viterbi even while controlling memory.

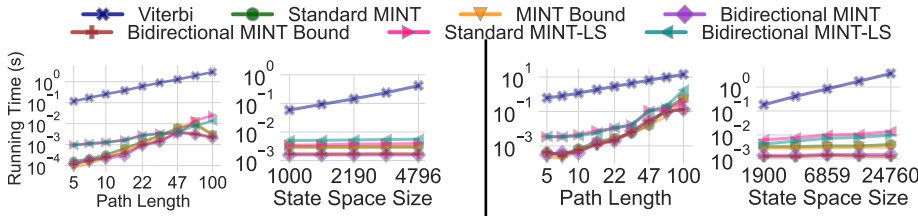

Figure 2: Decoding, real data; forced alignment (L), standard decoding (R); runtime vs. $T$ and $K$.

**Real data, runtime and memory vs. $T$.** Figure 3 plots runtime and the min, max, and median memory usage across recursion levels, including SIEVE variants. While DP baselines require *static* memory, MINT-LS's needs are *dynamic*, hence we consider its peak memory per level. The median memory needs of MINT-LS using DFS grow modestly with $T$; the maximum rises with $T$, yet remains under that of SIEVE-Middlepath, the most memory-thrifty SIEVE variant. We show results vs. $T$, as subsampling the state space on real data barely affects MINT-LS, as Figure 2 established.

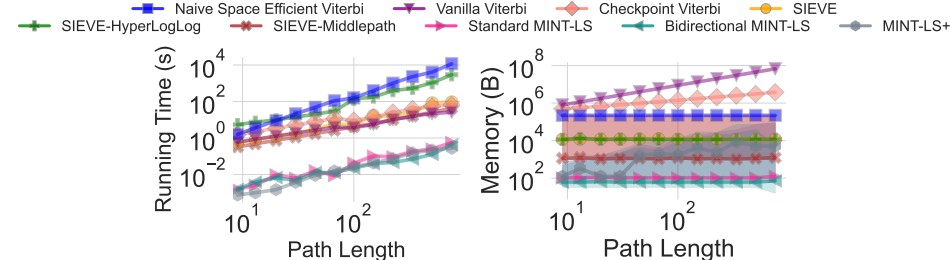

Figure 3: Decoding, real forced alignment data; runtime (L) and memory consumption (R) vs $T$. Shaded regions indicate the range of memory consumed over recursive calls; axes on log scale.

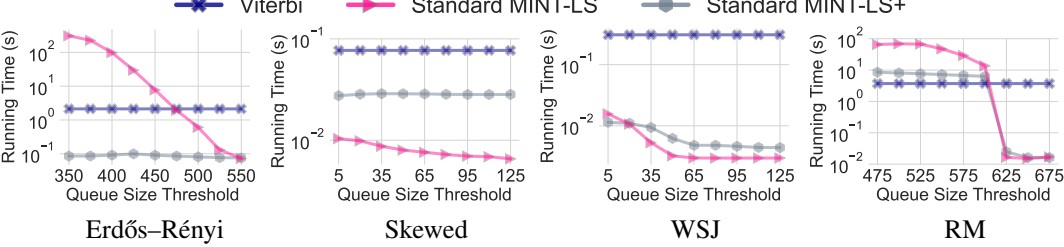

Figure 4: Decoding: synthetic data (Erdős–Rényi) with uniform and skewed path likelihood distribution (Skewed), and real data on forced alignment (WSJ) and standard decoding (RM). Runtime vs. queue size threshold $\theta$; $y$-axis in log scale.

**Effect of $\theta$ on linear-space algorithms.** Figure 4 presents runtimes vs. the queue size threshold $\theta$, focusing on ranges of the $\theta$ domain where MINT-LS may process tokens more than once, as explained in Section 3.1. MINT-LS+, on the other hand, never processes the same token more than once.

SIEVE (Ciaperoni et al., 2022), a DP-based rather than A*-based algorithm, ensures $\mathcal{O}(K)$ space complexity too, as it always stores $K$ active tokens. However, SIEVE incurs a practical runtime overhead *(not time-complexity overhead)* compared to Viterbi; we thus compare runtime to the latter as a baseline. MINT-LS consistently performs the fastest on synthetic data with skewed path likelihood distribution, where only a few paths are explored, yet becomes inefficient for small $\theta$ with uniform path likelihood distribution, where more paths are explored. On real data with forced alignment, the runtime of MINT-LS grows modestly as $\theta$ falls; with larger decoding data, runtime grows beyond that of Viterbi. Overall, MINT-LS is preferable for large enough $\theta$, while MINT-LS+ maintains runtime within the same order of magnitude as, and typically lower than, that of Viterbi.

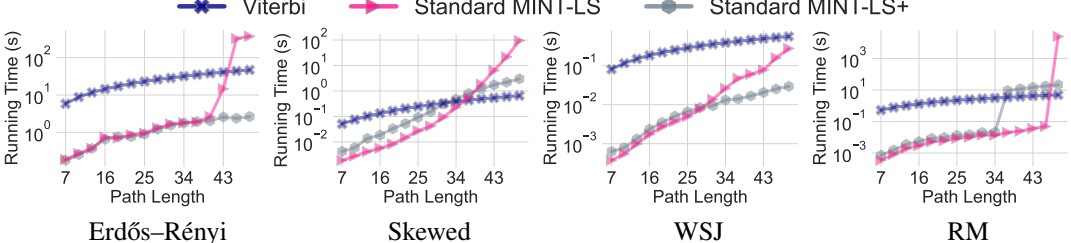

Figure 5: Decoding: synthetic data (Erdős–Rényi) with uniform and skewed path likelihood distribution (Skewed), and real data on forced alignment (WSJ) and standard decoding (RM). Runtime vs. path length $T$; $y$-axis in log scale.

**Effect of $T$ on linear-space algorithms.** Figure 5 plots the runtimes vs. path length $T$ for fixed values of the queue size threshold $\theta$ sufficiently small to reveal the challenges MINT-LS faces. The results reconfirm that MINT-LS becomes inefficient as more paths are explored. As in Figure 4, this effect is evident in synthetic data with Erdős–Rényi transitions and uniform path likelihood distribution and in the real decoding data. MINT-LS+ emerges as the algorithm of choice, achieving competitive runtime even while maintaining memory usage under a small $\theta$ threshold.

### 4.2 RESULTS ON HISTOGRAM CONSTRUCTION

Figure 6 shows the histogram construction runtime on synthetic data vs. $B$ for input sequence length $n = 1010$, and that of Standard TECH and the standard V-OPT algorithm vs. $n$ for different values of $\lambda = \frac{B}{n}$. Standard TECH often suffices, as its variants do not bring significant gains. Linear-space TECH incurs a manageable runtime overhead for the sake of space efficiency. All compact-A* variants gain in time efficiency as $B$ grows, delimiting the search space for each bucket. Contrariwise, standard DP cannot contain the search space, hence its runtime surges for large $B$. Figure 6 also shows results on real data. The gains of TECH solutions are *more emphatic* here, as TECH exploits data patterns that facilitate summarization, whereas standard DP lacks such capacity.

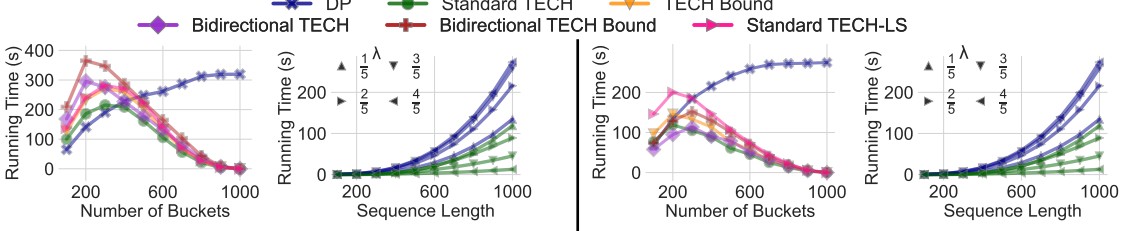

Figure 6: Histogram construction, synthetic (L) and real (R) data; runtime vs. $B$, $n$ varying $\lambda = \frac{B}{n}$.

## 5 CONCLUSION

We introduced compact-A*, a framework that efficiently solves problems of fixed-length path optimization by best-first search, while delimiting space complexity. We designed compact-A*-based algorithms for Viterbi decoding and histogram construction. In Appendix E, we apply compact-A* to another problem, temporal-graph community search. Our experiments evince that compact-A* gains up to four orders of magnitude in time and space efficiency, the advantage being more pronounced on real data with nonuniform path cost distributions.

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

## A  PROOFS

**Proposition 1.** *Standard MINT is correct.*

*Proof.* Let $\mathcal{T}$ be the set of all tokens (i.e., state-frame pairs) and $\mathbf{V}$ the set of visited tokens, for which optimal cost has been computed. Initially $\mathbf{V}$ includes the source $s$ with optimal cost $p_s = -\log B_{s,y_1}$, and $-p_s$ is its log-likelihood. In each iteration, we add to $\mathbf{V}$ a token $(s_i, t)$ from $\mathcal{T} \setminus \mathbf{V}$ with cost $p_i$. To complete the induction, we must show that $p_i$ is optimal for its length $t$. First, if the true $t$-length-optimal path goes only through tokens in $\mathbf{V}$, for which, by the inductive hypothesis, the optimal cost is known, then $p_i$ must be $t$-length-optimal. Assume that the $t$-length-optimal path goes through a token $(s_j, t')$ not in $\mathbf{V}$; this token necessarily has cost $p_j \geq p_i$, hence any path through it is suboptimal. Therefore, $-p_i$ is the maximum Viterbi path log-likelihood $\max_{s_i} \mathbf{T}[s_i, t]$.  $\square$

**Proposition 2.** *MINT Bound is correct.*

*Proof.* We prove the statement by contradiction. Assume the algorithm returns a token $(s^*, T)$ with cost $c^* > \max_{s_i} \mathbf{T}[s_i, T]$. Then there must be an unvisited token $(s', t')$, which, if propagated to the last frame, produces a path of likelihood $\max_{s_i} \mathbf{T}[s_i, T]$. Such a token would have priority $c' + (T - t') \cdot \hat{c}_1$ where $c'$ is the real cost of arriving to $s'$ in $t'$ frames. Since $\hat{c}_1$ lower-bounds the cost of moving from frame to frame, it follows that $c' + (T - t') \cdot \hat{c}_1 \leq \max_{s_i} \mathbf{T}[s_i, T] < c^*$ at any time $t' \leq T$. Therefore $(s^*, T)$ cannot be dequeued before $(s', t')$, ergo the proof is completed.  $\square$

**Proposition 3.** *Bidirectional MINT is correct.*

*Proof.* The algorithm alternates between a forward and backward step and maintains the best-so-far path of $T$ steps. Correctness rests on the stopping condition. The algorithm terminates when either (i) both queues are empty or (ii) the elements $(s_i^f, t^f)$ and $(s_i^b, t^b)$ popped from the queues have joint cost $d_f[(s_i^f, t^f)] + d_b[(s_i^b, t^b)]$ that is larger than the current best path cost $\mu$. Regarding condition (i), when both queues are empty, all possible paths have been generated, so the algorithm returns the optimal. Regarding condition (ii), assume that the optimal path is not yet found when the algorithm terminates returning $-\mu$. Then there must be a path $Q^*$ of cost $\mu^* < \mu$ containing at least one not yet visited token in $\mathbf{V}_f \cup \mathbf{V}_b$. Such a token would have cost at least $d_f[(s_i^f, t^f)]$ on the forward side and $d_b[(s_i^b, t^b)]$ on the backward side, hence path $Q^*$ would have cost $\mu^* \geq \mu$, a contradiction.  $\square$

**Proposition 4.** *The minimum error $\min_b E_b$ among $b$ buckets of an equal-width partitioning a sequence $I$ is a lower bound to the V-optimal histogram cost and $\sum_b E_b$ is an upper bound.*

*Proof.* Let $H_B$ be the V-optimal histogram of size $B$ on sequence $I$ and $H'_B$ be the histogram of the same number of buckets on the same sequence, where all buckets have the same size, except possibly the last. Let $[s_j, e_j]$ be the boundary positions and $E_j$ the error of the $j^{\text{th}}$ bucket in $H_B$, and let $[s'_j, e'_j]$ and $E'_j$ be the corresponding boundary positions and error of the $j^{\text{th}}$ bucket in $H'_B$. Then at least one bucket of $H_B$, say the $j^{\text{th}}$, has $a_j \leq a'_j$ and $b_j \geq b'_j$, i.e., it fully contains the corresponding bucket in $H'_B$. Then, as $E$ is monotonically non-decreasing with bucket width, $E_j \geq E'_j$; besides, $E_j \leq \sum_{h \in H_B} E_h$, ergo $\min_h E'_h \leq E'_j \leq E_j \leq \sum_{h \in H_B} E_h$, hence $\min_h E'_h$ is a lower bound to the V-optimal histogram cost. Furthermore, $\sum_h E'_h$ is an upper bound on the V-OPT histogram cost, since, by definition of the V-OPT histogram $H_B$, $\sum_h E_h \leq \sum_h E'_h$.  $\square$

## B  DATA, MEASURES, AND PARAMETERS

**Data.** We experiment on both synthetic and real-world datasets.

We evaluate MINT and TECH variants on synthetic data generated according to the following models:

- *Erdős–Rényi* model where each hidden state is emitting and connected with any other state with probability $p = 0.01$; transition and emission probabilities are generated uniformly at random, thus arbitrary cycles may be present. All states are emitting.

- *Skewed path likelihood* model, where we generate a fixed number $N_{path}$ (100, by default) of paths of length $T$ starting from initial state $s$, and composed of emitting states. To each such path we assign a probability drawn from a power law distribution $p(x, \alpha) = \alpha x^{\alpha-1}$, which we distribute evenly across transition and emission probabilities of all states in the path. As in the previous case, all states are emitting. We use this model to investigate how skew in the distribution of path likelihoods affects the advantage granted by compact-A*-based solutions.

We also assess MINT on real speech data:

- *Wall Street Journal (WSJ) corpus* data: we use a real-world composite HMM for speech-text forced alignment, the process of aligning text to audio recordings, which is also tackled by the Viterbi algorithm. The model is built using the HTK software toolkit Young et al. (2002) and contains 5529 states (including initial states), out of which 3204 are emitting; it was trained on the WSJ corpus Paul & Baker (1992) aiming to align speech recordings from the TIMIT corpus Garofolo et al. (1993).

- *Resource Management (RM) corpus* data: we use a real-world composite HMM for decoding, trained on the RM speech corpus Price et al. (1993) and built using the Kaldi software toolkit Povey et al. (2011), The graph comprises 25 333 states (including initial state) and 175 428 edges, out of which 162 255 also carry emission probabilities. We decode subsets of a simple recorded utterance of up to 100 frames.

The observation sequence $Y$ consists of feature vectors of Mel-Cepstrum cepstral coefficients and their derivatives and emission probabilities are given by multivariate Gaussian mixture models.

Similarly, we evaluate TECH in (i) synthetic *sequences of integers* in the range $[0, 50]$ and (ii) *Dow-Jones Industrial Average (DJIA) closing values* real data.

**Metrics.** We measure runtime in seconds (s) and memory in bytes (B). In all cases, we report averages over 5 runs.

**Parameters.** Regarding decoding in HMMs, in experiments with synthetic data, we vary $T$ in geometric progression of 9 values from 5 to 30 with $K = 7500$; in experiments with real data, we vary $T$ in geometric progression of 9 values from 5 to 100 with $K$ fixed to the size of the original state space. Furthermore, in synthetic data we vary $K$ in a geometric progression of 5 values from 1000 to 16000 with $T = 10$; in real data, we vary $K$ over 5 values in geometric progression from 1000 (forced alignment) or 1900 (decoding) to approximately the size of the original state space, with $T = 30$. To vary $K$ in real-world HMMs, we sample subsets of the original HMM graph via snowball sampling from a start state $source$. To investigate the combined impact of $K$ and $T$, we also vary them simultaneously. We also vary $K \in \{25 \times 10^3, 30 \times 10^3\}$ and $T \in \{25, 30\}$ to monitor how MINT variants behave during runtime in terms of memory usage and the evolution of path likelihood. In the experiment with the skewed path likelihood model, we also vary the power law parameter $\alpha$ controlling skewness in 17 values from $10^{-2}$ to $10^2$ with $N_{path} = 100$, $T = 10$ and $K = 1500$. In the space-efficient standard and bidirectional MINT-LS, unless specified otherwise, we set the queue size threshold $\theta$ to 10% of $K \times T$. However, with Erdős–Rényi data, which call for a larger budget as they reflect a worst-case scenario, we set $\theta$ to 90% of $K \times T$. In MINT-LS+, unless specified otherwise, we set $\theta = K$.

To assess the runtime of MINT-LS to MINT-LS+ as a function of the queue size threshold $\theta$, we use synthetic data with $K = 7500$ and $T = 10$ and real data with $K$ defined by the state space and $T = 30$, varying $\theta$ in arithmetic progression with step 15 from 5 to 125 on synthetic data with skewed path likelihood distribution and real forced alignment data, where the amounts of paths to be explored is limited, from 350 to 550 on Erdős–Rényi data, and from 475 to 675 on real decoding data. We set the $\Delta_Q$ in MINT-LS+ to 2. To assess the runtime of MINT-LS to MINT-LS+ as a function of path length $T$, we use synthetic data with $K = 16000$ and real data with $K$ defined by the state space, varying $T$ in arithmetic progression with step 3 from 7 to 49. We set $\theta = 2700$ on synthetic Erdős–Rényi data, $\theta = 10$ on synthetic data with skewed path likelihood distribution, $\theta = 20$ on real forced alignment data, and $\theta = 800$ on real decoding data, and $\Delta_Q$ in MINT-LS+ to 5.

In histogram construction experiments, we vary $B$ from 100 to 1000, while holding $n$ fixed to 1010. We also consider sequences of length $n$ increasing from 100 to 1000 for the different values of $\lambda = \frac{B}{n}$ indicated in the results. With TECH-LS, we set the queue size threshold to 10% of $n \times B$.

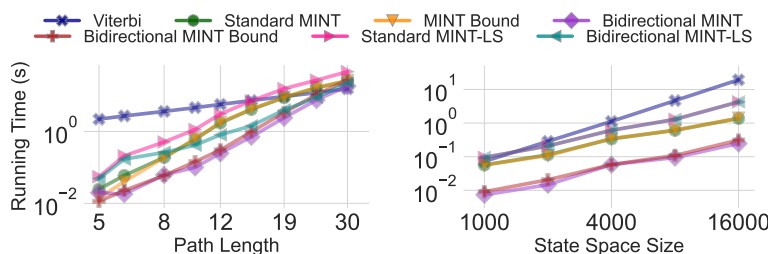

Figure 7: Decoding, synthetic data. Runtime vs. path length (left) and state space size (right).

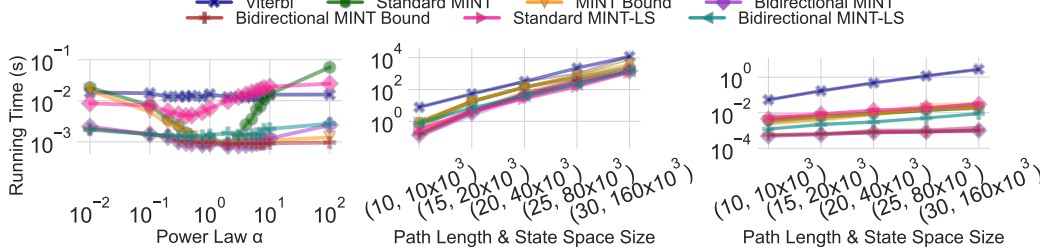

Figure 8: Decoding; synthetic data with skewed path likelihoods; runtime vs. skew $\alpha$ (left); synthetic Erdős–Rényi data (center) and synthetic data with skewed path likelihoods (right): runtime vs. both linearly growing path length $T$ and exponentially growing state space size $K$, indicated as $(T, K)$; shaded regions indicate the minimum and maximum runtime due to randomness in data generation; both axes on log scale (left) and $y$-axis in log scale (center and right).

## C  ADDITIONAL EXPERIMENTS

**Synthetic data, runtime vs. $T$ and $K$.** Figure 7 plots runtime vs. path length $T$ and state space size $K$ on Erdős–Rényi data. MINT accelerates decoding significantly, thanks to visiting only a few tokens. MINT-LS incurs marginal runtime overhead compared to MINT. Still, processing a single token is faster in Viterbi, as MINT's queue management incurs overhead, causing savings to drop as $T$ grows. This result stems from the data model, which reflects a worst-case scenario whereby path likelihoods converge for large enough $T$, leading MINT to visit too many tokens. Conversely, in real-world speech data with probabilities concentrated over a limited subset of paths, MINT yields higher savings by focusing on the most promising paths.

**Runtime vs. $\alpha$.** To demonstrate the effect of path likelihood skew using synthetic data, Figure 8 plots runtime as a function of the parameter $\alpha$ of the power law distribution over the path likelihoods. Notably, the highest savings are obtained for $\alpha$ close to 1. This is due to the fact that, for remarkably smaller or larger $\alpha$, all paths tend to be equally likely, so MINT cannot focus on a small subset of paths. Nevertheless, even in the case where the path likelihood distribution approaches the worst case, as in the Erdős–Rényi model, we still have high savings for small $T$, which is a popular setting in modern speech recognition. Regarding different implementations of MINT, we observe that the use of lower bounds is not always crucial for runtime; however, as $\alpha$ increases, MINT Bound vastly outperforms Standard MINT by virtue of its capacity to prune paths from consideration more aggressively. Furthermore, bidirectional-search variants accomplish the highest efficiency on synthetic data, both those generated by the Erdős–Rényi model and those with skewed path likelihoods; these results vindicate our development of those enhanced solutions.

**Runtime vs. $T$ and $K$ tuned in unison.** We also measure runtime as a function of both $T$ and $K$ on Erdős–Rényi model data and on those with skewed path likelihood distribution with $\alpha = 1$. Figure 8 shows the results. Shaded regions indicate the minimum and the maximum over runs, which convey the extent of random variation; as the figure shows, that extent is quite limited. The savings observed as we increase both $T$ and $K$ are consistent with our previous findings and most pronounced in the skewed likelihoods scenario. In the Erdős–Rényi model, as most paths of a given length have similar likelihoods, the savings are more modest and decrease with the growth of both $T$ and $K$.

**Real-time memory monitoring.** Figure 9 shows memory requirements at run time for four parameter configurations on synthetic data with skewed path likelihood distribution using $\alpha = 1$; for reference, we also provide the constant memory used by standard Viterbi. Notably, MINT variants reduce the memory requirements of Viterbi by several orders of magnitude. Unsurprisingly, the two bidirectional-search variants consume slightly more memory, yet need fewer iterations till termination, as they apply both a forward and a backward search with two queues. With MINT-LS, we show memory consumption vs. iterations or DFS calls. While by the chosen budget, MINT-LS variants use as little as $25\%$ of the memory used by MINT variants; this advantage may grow on demand by reducing the queue size threshold $\theta$.

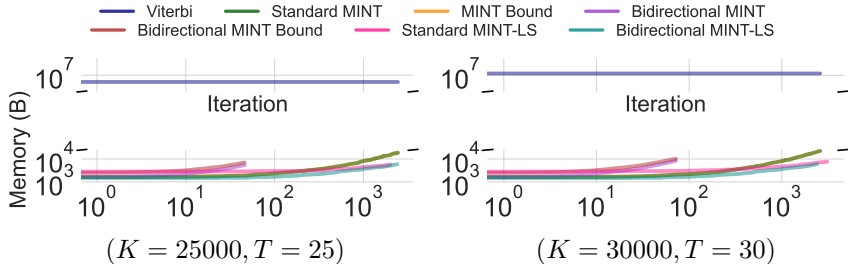

Figure 9: Synthetic data with skewed path likelihood distribution; memory requirements on the fly.

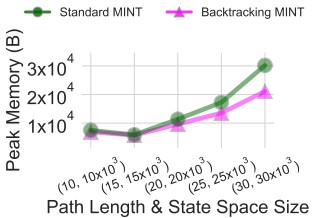

Figure 10: Decoding, synthetic data with skewed path likelihood distribution. Memory requirements of default MINT and memory-efficient MINT-Backtracking. Maximum (peak) memory vs. path length $T$ and state space size $K$, indicated as $(T, K)$; $\frac{K}{T}$ is fixed.

**Effect of backtracking.** As explained in Section 3.1, by default MINT variants store paths explicitly; however, we may reduce memory usage by only storing the predecessor of each token $(s_i, t)$ and eventually reconstructing the optimal path by backtracking over such links, with a small runtime overhead and savings in memory consumption. We refer to the resulting implementation as MINT-Backtracking. To illustrate this effect, Figure 10 presents the maximum memory usage of the two implementations of standard MINT under the HMM graph model with skewed path likelihood distribution ($\alpha = 1$) as a function of both $K$ and $T$. While the difference in memory requirement is evident, we measured the corresponding runtime difference to be negligible. MINT-LS employs the backtracking implementation for the sake of space efficiency. Besides, MINT-Backtracking extends seamlessly to all variants of MINT. In the case of bidirectional-search-based variants, backtracking proceeds in both directions after the optimal path is found.

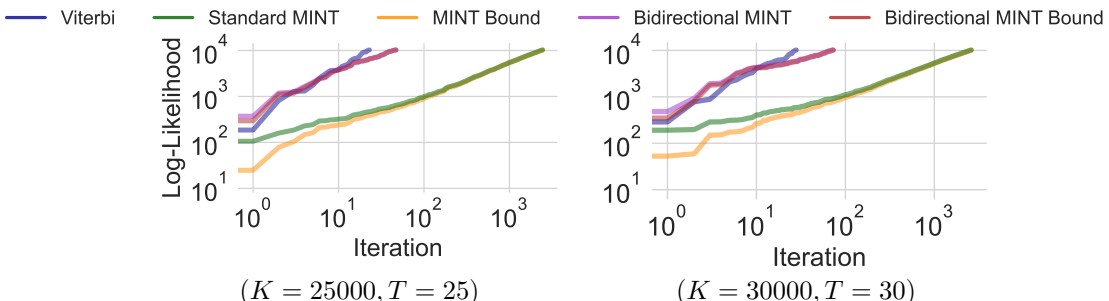

$(K = 25000, T = 25)$ $\qquad\qquad$ $(K = 30000, T = 30)$

Figure 11: Synthetic data, skewed path likelihood distribution; path log-likelihood (in absolute value) on the fly; axes on log-scale.

**Real-time log-likelihood monitoring.** We also monitor the optimal path log-likelihood absolute value across iterations. This *absolute* value grows, as longer paths have *lower* likelihood than shorter

ones. Figure 11 presents our results, using the same four parameter configurations as in Figure 9. In all algorithms the likelihood approaches the optimal value swiftly and monotonically. In the case of standard Viterbi, we plot the highest likelihood found at the end of each frame (i.e., path length considered), hence Viterbi appears to undergo fewer iterations. For the two bidirectional MINT variants, we plot the sum of likelihoods associated with the last tokens de-queued from the forward queue $\mathbf{Q}_f$ and the backward queue $\mathbf{Q}_b$ in each iteration. For other algorithms, we plot the likelihood associated with the token dequed from $\mathbf{Q}$ in each iteration. We found that MINT-LS variants exhibit the same progression of path likelihood as the corresponding MINT variants.

---

**Algorithm 1:** Standard MINT

**Data:** HMM graph $G$, transition and emission probabilities $A$ and $B$, observations $Y$, and initial state $s$.
**Result:** Viterbi Path Log-Likelihood $\max_{s_i} \mathbf{T}[s_i, T]$.

1  $\mathbf{Q} \leftarrow Queue((s, 1), p(s, 1) = -\log B_{s,y_1})$;
2  $\mathbf{V} \leftarrow \{\}$;
3  **while** $\mathbf{Q} \neq \emptyset$ **do**
4      $(s_i, t), p_i \leftarrow \mathbf{Q}.\text{pop}()$; // (state, frame), priority
5      **if** $t = T$ **then break**;
6      $\mathbf{V}.\text{add}((s_i, t))$;
7      **for** $s_j$ *in* $G[s_i]$ **do**
8          **if** $(s_j, t+1) \notin \mathbf{V}$ **then**
9              $d \leftarrow p_i - \log A_{s_i,s_j} - \log B_{s_j,y_{t+1}}$;
10             **if** $(s_j, t+1) \notin \mathbf{Q}$ **then** $\mathbf{Q}.\text{insert}((s_j, t+1), p(s_j, t+1) = d)$ ;
11             **if** $\mathbf{Q}[(s_j, t+1)] > d$ **then** $\mathbf{Q}.\text{update}((s_j, t+1), p(s_j, t+1) = d)$ ;
12 **return** $-p_i$;

---

**Algorithm 2:** Bidirectional MINT

**Data:** HMM graph $G$, transition and emission probabilities $A$ and $B$, observations $Y$, and initial and final states $source$ and $target$.
**Result:** Viterbi Path Log-Likelihood $\max_{s_i} \mathbf{T}[s_i, T]$.

1  $\mathbf{Q}_f \leftarrow Queue((source, 1), p(source, 1) = -\log B_{source,y_1})$;
2  $\mathbf{Q}_b \leftarrow Queue((target, T), p(target, T) = 0)$;
3  $\mathbf{V}_f \leftarrow \{\}; \mathbf{V}_b \leftarrow \{\}; \mu \leftarrow \infty$;
4  **while** $\mathbf{Q}_b \neq \emptyset \wedge \mathbf{Q}_f \neq \emptyset$ **do**
5      $(s_i^f, t^f), p_i^f \leftarrow \mathbf{Q}_f.\text{pop}(); (s_i^b, t^b), p_i^b \leftarrow \mathbf{Q}_b.\text{pop}()$;
6      $d_f[s_i^f] \leftarrow p_i^f; d_b[s_i^b] \leftarrow p_i^b$;
7      $\mathbf{V}_f.\text{add}((s_i^f, t^f)); \mathbf{V}_b.\text{add}((s_i^b, t^b))$;
8      **if** $t^f < T$ **then**
9          **for** $s_j$ *in* $G[s_i^f]$ **do**
10             Update $\mathbf{Q}_f$ for $(s_j, t+1)$;
11             **if** $(s_j, t+1) \in \mathbf{V}_b \wedge d_f[(s_i^f, t^f)] - \log A_{s_i^f, s_j} - \log B_{s_j, y_{t+1}} + d_b[(s_j, t+1)] < \mu$ **then**
12                 $\mu = d_f[(s_i^f, t^f)] - \log A_{s_i^f, s_j} - \log B_{s_j, y_{t+1}} + d_b[(s_j, t+1)]$;
13     **if** $t^b > 1$ **then**
14         **for** $s_j$ *in* $G_{rev}[s_i^b]$ **do**
15             Update $\mathbf{Q}_b$ for $(s_j, t-1)$;
16             **if** $(s_j, t-1) \in \mathbf{V}_f \wedge d_b[(s_i^b, t^b)] - \log A_{s_j, s_i^b} - \log B_{s_i^b, y_t} + d_f[(s_j, t-1)] < \mu$ **then**
17                 $\mu = d_b[(s_i^b, t^b)] - \log A_{s_j, s_i^b} - \log B_{s_i^b, y_t} + d_f[(s_j, t-1)]$;
18     **if** $d_f[(s_i^f, t^f)] + d_b[(s_i^b, t^b)] \geq \mu$ **then break**;
19 **return** $-\mu$;

---

## D    Pseudocodes

Here we collect pseudocodes for the presented algorithms.

Algorithm 1 presents MINT. Algorithm 2 presents Bidirectional MINT, assuming that both an initial and final state are given. Algorithm 3 illustrates TECH. Algorithm 5 provides the pseudocode of MINT-LS, which uses Algorithm 4 as a subroutine. Lastly, Algorithm 6 presents MINT-LS+.

## E    Case Study

As a case study, we apply compact-A* to the problem of *temporal community search* Galimberti et al. (2021). Given a temporal graph $G_{\mathcal{T}}$ over a temporal domain $\mathcal{T} = [0, 1, \ldots t_{max}]$, an integer $h$, and a set of query nodes $q$, the problem seeks a partitioning $P$ of the temporal domain into $h$ segments and a subgraph $G_h$ that contains the query nodes $q$ within each bucket and maximizes the *sum of minimum degrees* of subgraphs in $P$. This problem is pertinent as the growing availability of timestamped

**Algorithm 3:** Standard TECH

**Data:** Input sequence $I$, integer $B$.
**Result:** V-Optimal Histogram Error $E^*(n, B)$.

1  $\mathbf{Q} \leftarrow Queue((1, 1), p(1, 1) = 0)$;
2  $S \leftarrow []; SS \leftarrow []; \mathbf{V} \leftarrow \{\}; n \leftarrow I.length$;
3  $S[1] \leftarrow I[1]; SS[1] \leftarrow I[1]^2$;
4  **for** $i \in \{2, \ldots, n\}$ **do**
5  $\quad$ $S[i] \leftarrow S[i-1] + I[i]; SS[i] \leftarrow SS[i-1] + I[i]^2$;
6  **for** $j \in \{2, \ldots, n - (B-1)\}$ **do**
7  $\quad$ $\mathbf{Q}.insert((j, 1), p(j, 1) = E(1, j))$;
8  **while** $\mathbf{Q} \neq \emptyset$ **do**
9  $\quad$ $(i, b), p \leftarrow \mathbf{Q}.pop()$; // (values,buckets),priority
10  $\quad$ $\mathbf{V}.add((i, b))$;
11  $\quad$ **if** $b = B \wedge i = n$ **then break**;
12  $\quad$ **if** $b < B$ **then**
13  $\quad\quad$ **for** $j \in \{i+1, \ldots, n - B + b + 1\}$ **do**
14  $\quad\quad\quad$ **if** $(j, b+1) \notin \mathbf{V}$ **then**
15  $\quad\quad\quad\quad$ $d \leftarrow p + E(i, j)$;
16  $\quad\quad\quad\quad$ **if** $(j, b+1) \notin \mathbf{Q}$ **then**
17  $\quad\quad\quad\quad\quad$ $\mathbf{Q}.insert((j, b+1), p(j, b+1) = p + E(i+1, j))$
18  $\quad\quad\quad\quad$ **if** $\mathbf{Q}[(j, b+1)] \geq p + E(i+1, j)$ **then**
19  $\quad\quad\quad\quad\quad$ $\mathbf{Q}.update((j, b+1), p(j, b+1) = p + E(i+1, j))$
20  **return** $p$;

---

**Algorithm 4:** DFS

**Data:** HMM graph $G$, transition and emission probabilities $A$ and $B$, states $S$, observations $Y$, initial state $s_i$, predecessor pred, initial frame $t$, initial path priority $p_i$, middle pair, queue $\mathbf{Q}$, middle frame.
**Result:** updated queue $\mathbf{Q}$.

1  **if** $t = $ middle_frame **then**
2  $\quad$ middle_pair $\leftarrow (pred, s_i)$; // update middle pair
3  **if** $t < T$ **then**
4  $\quad$ **for** $s_j$ *in* $G[s_i]$ **do**
5  $\quad\quad$ **if** $s_j \in S$ **then**
6  $\quad\quad\quad$ $d \leftarrow p_i - \log A_{s_i, s_j} - \log B_{s_j, y_{t+1}}$;
7  $\quad\quad\quad$ **if** $(s_j, t+1) \in \mathbf{Q}$ **then**
8  $\quad\quad\quad\quad$ **if** $\mathbf{Q}[(s_j, t+1)] > d$ **then**
9  $\quad\quad\quad\quad\quad$ $\mathbf{Q}.update((s_j, t+1), p(s_j, t+1) = d, pred = s_i, \text{middle\_pair} = \text{middle\_pair})$;
10  $\quad\quad\quad$ **else**
11  $\quad\quad\quad\quad$ $\mathbf{Q} \leftarrow$ DFS$(G, A, B, S, Y, s_j, s_i, t+1, d, \text{middle\_pair}, \mathbf{Q}, \text{middle\_frame})$; // continue DFS
12  **else**
13  $\quad$ **if** $(s_i, t) \notin \mathbf{Q}$ **then** $\mathbf{Q}.insert((s_i, t), p(s_i, t) = p_i, pred = pred, \text{middle\_pair} = \text{middle\_pair})$ ;
14  $\quad$ **else** $\mathbf{Q}.update((s_i, t), p(s_i, t) = p_i, pred, \text{middle\_pair})$;
15  **return** $\mathbf{Q}$;

---

data generates interest in temporal graph management. Real-world temporal graphs typically align themselves in evolving communities, which one may study by focusing on a set of query nodes.

The problem is solved by the DP recursion:

$$p^*(i, b) = \max_{0 \leq j < t_{max}} p^*(j, b-1) + v_q^*(j+1, i), \tag{5}$$

where $p^*(i, b)$ denotes the optimal objective value for a partition of the first $i$ timestamps in $b$ segments and $v_q^*(j+1, i)$ is the maximum *minimum degree* of a subgraph containing query nodes $q$ and enduring from the $(j+1)^{\text{th}}$ to the $i^{\text{th}}$ timestamp. A cross-examination of Equations (5) and (4) reveals their analogy, with the main difference lying in the value associated with each segment, i.e., in the terminology of Section 3, the *gap function*. Thus, the dynamic-programming algorithm for histogram construction also solves temporal community search with the necessary modifications.

Nevertheless, to compute the gap function $v_q^*(j, i)$ we need to identify a subgraph containing the query nodes $q$ of maximum minimum degree, for each query and each of the $O(t_{max}^2)$ possible $(j, i)$-buckets. The solution in Galimberti et al. (2021) precomputes all gap function values through *span-core decomposition* and uses them in the dynamic-programming recursion of Equation (5).

We apply compact-A* to obtain an advantage over the DP solution to temporal community search on a real-world temporal network that captures interactions between students and teachers of nine high school classes in France over five days (Mastrandrea et al., 2015). The temporal graph has 47.590 edges (interactions) and 327 nodes (students and teachers). These parameters only affect the offline

**Algorithm 5: MINT-LS**

---

**Data:** HMM graph $G$, transition and emission probabilities $A$ and $B$, states $S$, observations $Y$, queue size threshold $\theta$, initial and final state startSt and lastSt, initial and final frame startFr and lastFr.

1   middle_frame $\leftarrow \lceil (\text{startFr} + \text{lastFr})/2 \rceil$, $\mathbf{V} \leftarrow \{\}$ // initialization
2   $\mathbf{Q} \leftarrow Queue((\text{startSt}, \text{startFr}), p(\text{startSt}, \text{startFr}) = d, \text{pred} = -1, \text{middle\_pair} = (-1, -1));$
3   **while** $\mathbf{Q} \neq \emptyset$ **do**
4      $(s_i, t), p_i, \text{pred}, \text{middle\_pair} \leftarrow \mathbf{Q}.\text{pop}();$
5      $\mathbf{V}.\text{add}((s_i, t));$
6      **if** $t = \text{middle\_frame} \wedge \text{middle\_pair} = (-1, -1)$ **then**
7         middle_pair $\leftarrow (\text{pred}, s_i)$; // update middle pair
8      **if** $(t = \text{lastFr} \wedge \text{lastSt} = -1) \vee (t = \text{lastFr} \wedge \text{lastSt} = s_i)$ // lastSt $= -1$ if not input
9      **then**
10         $s_{m-}, s_{m+} \leftarrow$ middle_pair; // extract middle pair
11         $N_p \leftarrow$ middle_frame; // number of frames before the middle pair
12         **if** $N_p > 1$ // continue recursion in predecessors
13         **then**
14            $S_p \leftarrow$ FIND-T-HOPPRED$(s_{m-}, N_p)$; // find predecessors of $s_{m-}$
15            MINT-LS$(G, A, B, S_p, Y, F, \theta, \text{startSt}, s_{m-}, \text{startFr}, N_p)$;
16         $N_s \leftarrow \text{startFr} + N_p$; // number of frames after the middle pair
17         **print** $(s_{m-}, s_{m+})$; // in-order print of middle pairs
18         **if** $N_s > 1$ // continue recursion in successors
19         **then**
20            $S_s \leftarrow$ FIND-T-HOPSUCC$(s_{m+}, N_s)$; // find successors of $s_{m+}$
21            MINT-LS$(G, A, B, S_s, Y, F, \theta, s_{m+}, \text{lastSt}, N_s, \text{lastFr})$;
22      **for** $s_j$ *in* $G[s_i]$ **do**
23         **if** $(s_j, t+1) \notin \mathbf{V} \wedge s_j \in S$ **then**
24            $d \leftarrow p_i - \log A_{s_i, s_j} - \log B_{s_j, y_{t+1}}$;
25            **if** $\mathbf{Q}[(s_j, t+1)] > d \vee (s_j, t+1) \notin \mathbf{Q}$ **then**
26               **if** $\mathbf{Q}.\text{size}() > \theta \wedge (s_j, t+1) \notin \mathbf{Q}$ **then**
27                  DFS$(G, A, B, S, Y, s_j, \text{pred}, t+1, d, \text{middle\_pair}, \mathbf{Q}, \text{middle\_frame})$;
28               **else**
29                  **if** $(s_j, t+1) \notin \mathbf{Q}$ **then**
                     $\mathbf{Q}.\text{insert}((s_j, t+1), p(s_j, t+1) = d, \text{pred} = s_i, \text{middle\_pair} = \text{middle\_pair})$;
30                  **else** $\mathbf{Q}.\text{update}((s_j, t+1), p(s_j, t+1) = d, \text{pred} = s_i, \text{middle\_pair} = \text{middle\_pair})$;

---

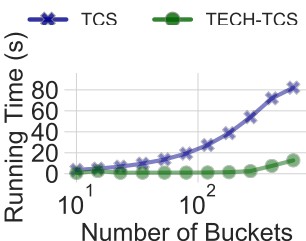

Figure 12: Temporal community search; runtime by number of buckets partitioning the domain.

pre-computation of gap function values and not the query processing phase. The length of the sequence to be partitioned is $t_{max} = 1212$. We apply the standard DP algorithm (TCS) and an algorithm based on Standard TECH (TECH-TCS) on the problem with a query comprising the node labelled 1. Figure 12 plots runtime vs. the number of buckets that partition the temporal domain, varied in geometric progression with ratio 1.5, from 10 to 608. Notably, TECH-TCS outpaces TCS, even for a few buckets. We obtained similar results with different query nodes and larger query node sets, as the query does not affect the search space of the DP solution and its compact-A* counterpart.

**Algorithm 6:** MINT-LS+

**Data:** HMM graph $G$, transition and emission probabilities $A$ and $B$, states $S$, observations $Y$, queue size threshold $\theta$, initial and final state startSt and lastSt, initial and final frame startFr and lastFr, maximum number of frames in the queue $\Delta_{\mathbf{Q}}$.

```
1  middle_frame ← ⌈(startFr + lastFr)/2⌉, V ← {};
   // initialization
2  Q ← Queue((startSt, startFr), p(startSt, startFr) = 0, pred = −1, middle_pair = (−1, −1));
3  Qₜ ← Queue((startSt, startFr), p(startSt, startFr) = startFr, cost = 0, pred = −1, middle_pair = (−1, −1));
4  current_max_frame ← startFr, EFS_flag ← False;
5  while Q ≠ ∅ do
6      if EFS_flag = True then
7          (sᵢ, t), t, pᵢ, pred, middle_pair ← Qₜ.pop();
8          Q.delete(((sᵢ, t));
9      else
10         (sᵢ, t), pᵢ, pred, middle_pair ← Q.pop();
11         Qₜ.delete((sᵢ, t));
12     V.add((sᵢ, t));
13     if t = middle_frame ∧ middle_pair = (−1, −1) then
14         middle_pair ← (pred, sᵢ); // update middle pair
15     if (t = lastFr ∧ lastSt = −1) ∨ (t = lastFr ∧ lastSt = sᵢ) // lastSt = −1 if not input
16     then
17         s_{m−}, s_{m+} ← middle_pair; // extract middle pair
18         Nₚ ← middle_frame; // # frames before middle
19         if Nₚ > 1 // continue recursion in predecessors
20         then
21             Sₚ ← FIND-t-HOPPRED(s_{m−}, Nₚ); // find predecessors of s_{m−}
22             MINT-LS+(G, A, B, Sₚ, Y, F, θ, startSt, s_{m−}, startFr, Nₚ, Δ_Q);
23         Nₛ ← startFr + Nₚ; // number of frames after the middle pair
24         print (s_{m−}, s_{m+}); // in-order print of middle pairs
25         if Nₛ > 1 // continue recursion in successors
26         then
27             Sₛ ← FIND-t-HOPSUCC(s_{m+}, Nₛ); // find successors of s_{m+}
28             MINT-LS+(G, A, B, Sₛ, Y, F, θ, s_{m+}, lastSt, Nₛ, lastFr, Δ_Q);
29     for sⱼ in G[sᵢ] do
30         if (sⱼ, t + 1) ∉ V ∧ sⱼ ∈ S then
31             d ← pᵢ − log A_{sᵢ,sⱼ} − log B_{sⱼ,y_{t+1}};
32             if Q[(sⱼ, t + 1)] > d ∨ (sⱼ, t + 1) ∉ Q then
33                 if (sⱼ, t + 1) ∉ Q then
34                     Q.insert((sⱼ, t + 1), p(sⱼ, t + 1) = d, pred = sᵢ, middle_pair = middle_pair);
35                     Qₜ.insert((sⱼ, t + 1), p(sⱼ, t + 1) = t + 1, cost = d, pred = sᵢ, middle_pair = middle_pair);
36                     if t + 1 > current_max_frame then
37                         current_max_frame ← t + 1; // update maximum frame in the queue
38                 else
39                     Q.update((sⱼ, t + 1), p(sⱼ, t + 1) = d, pred = sᵢ, middle_pair = middle_pair);
40                     Qₜ.update((sⱼ, t + 1), p(sⱼ, t + 1) = t + 1, cost = p, pred = sᵢ, middle_pair = middle_pair);
41     if EFS_flag = False then
42         if Q.size() > θ then
43             EFS_flag ← True; // activate EFS
44     else
45         (current_min_state, current_min_frame) ← Qₜ.top(); // access minimum frame in the queue
46         if Q.size() < θ ∨ current_max_frame − current_min_frame < Δ_Q then
47             EFS_flag ← False; // deactivate EFS
```

