# OpenReview forum: "Compact-A∗: Space-Efficient Fixed-Length Path Optimization"
_ICLR.cc/2026/Conference — ICLR 2026 Conference Withdrawn Submission_

### Official Review · Reviewer_ADPJ · 2025-10-25

**Soundness:** 3
**Presentation:** 3
**Contribution:** 2
**Rating:** 2
**Confidence:** 4

**Summary:**

This paper addresses the problem of finding an optimal sequence of a fixed length $L$ over a state space of size $n$, a task common in applications such as Viterbi decoding for Hidden Markov Models and V-optimal histogram segmentation.The authors highlight limitations in current standard approaches, specifically regarding the space-time trade-off. While Dynamic Programming (DP) and A* search are conventionally used, they often require $O(nL)$ space to memoize partial solutions for backtracking. Existing methods to reduce space to $O(n)$ either incur severe time overheads (e.g., $O(n^2L^2)$ by re-running algorithms) or rely on exhaustive breadth-first evaluations that limit scalability. Standard A* also struggles in this fixed-length context, as its priority queue and memoization needs grow to $O(nL)$ space.To overcome these bottlenecks, the paper introduces "Compact-A*", a framework designed for space- and time-efficient optimization of these fixed-length sequences. The key technical contributions involve using novel search strategies to control priority queue size and employing a divide-and-conquer approach to avoid traditional memoization, thereby achieving $O(n)$ space complexity. The authors validate Compact-A* on Viterbi decoding and V-optimal segmentation tasks, claiming improvements in both memory usage and runtime compared to prior work, particularly when dealing with skewed cost distributions.

**Strengths:**

Interesting and novel improvements over the A* algorithm, along with non-trivial experimental results.

**Weaknesses:**

While this paper presents an interesting topic, I believe it is not a strong fit for the ICLR venue. The work does not appear to focus on learning representations or deep learning/generative AI approaches, which are central themes for the ICLR community. Furthermore, statements such as "In modern speech-recognition systems, the Viterbi algorithm runs..." suggest a disconnect from current state-of-the-art practices. Modern speech recognition pipelines are predominantly end-to-end deep neural networks, and traditional methods like Viterbi are no longer considered "modern" in this context. This reinforces my impression that the paper may be better suited for a different venue.

**Questions:**

No questions

---

> ### Author Response · Authors · 2025-11-12
> **The Viterbi algorithm remains relevant to modern speech recognition pipelines**
>
> Thank you for the thorough review.
>
> >The work does not appear to focus on learning representations
>
> The sequence segmentation or histogram construction problem constructs an approximate representation of any signal or sequence.
>
> >Modern speech recognition pipelines are predominantly end-to-end deep neural networks, and traditional methods like Viterbi are no longer considered "modern" in this context.
>
> The Viterbi algorithm remains relevant to modern speech recognition pipelines, as indicated, for example, in [Pratap et al.: Scaling Speech Technology to 1,000+ Languages. JMLR 25 (2024) 1-52]. This 2024 paper explicitly mentions using the Viterbi algorithm and, moreover, reports facing a memory bottleneck when doing so, as "storing all $O(T × L)$ forward values for the Viterbi algorithm is infeasible on GPUs due to memory constraints". Our work address exactly this bottleneck.

---

### Official Review · Reviewer_xNKb · 2025-10-29

**Soundness:** 2
**Presentation:** 1
**Contribution:** 2
**Rating:** 2
**Confidence:** 2

**Summary:**

The authors introduce the compact-A* algorithm which space-efficiently addresses the fixed-length path optimization problems. They evaluate the performance of compact-A*-based implementations with several tasks such as Viterbi decoding and histogram construction.

The authors attempt to address challenging problems in computer science but the current paper has a couple of issues.

First, the topic does not match the subjects of ICLR well. In my understanding, although ICLR accepts papers on optimization, the authors' optimization is about operations research/heuristic search and not about optimization on learning and/or representation.

Second, the paper is very poorly written and hard to understand. For example, there are no concrete examples that clearly illustrate the problems as well how their algorithm behaves with step-by-step traces. Figure 1 does not help understand the problem definitions and the algorithmic behavior, because how it does not describe how each value in each square is calculated and it does not describe how calculations of the values are omitted by compact A* (we need to define a heuristic function and show a trace of the priority queue). As a result, I cannot accurately assess the technical novelty and correctness of their approach.

Finally, if I understand correctly, their divide-and-conquer strategy is an existing idea studied in solving the multiple sequence alignment problem. For example, Hirschberg's algorithm is the DP version, and Korf investigated both unidirectional and bidirectional A*.
I do not list all of the papers but there are other follow-up papers to these approaches in the heuristic search community.

1. Hirschberg, D.S., A linear space algorithm for computing maximal common subsequences, Communications of the ACM, Vol. 18, No. 6, June, 1975, pp. 341-343.
https://dl.acm.org/doi/10.1145/360825.360861

2. Richard E. Korf. Divide-and-Conquer Bidirectional Search: First Results. IJCAI 1999.
https://www.ijcai.org/Proceedings/99-2/Papers/073.pdf

4. Richard E. Korf and Weixiong Zhang. Divide-and-Conquer Frontier Search Applied to Optimal Sequence Alignment. AAAI 2000.
https://cdn.aaai.org/AAAI/2000/AAAI00-140.pdf

**Strengths:**

Attempt to address the issue in search and discrete optimization

Attempt to evaluate the performance of compact-A*-based implementations against other existing approaches

**Weaknesses:**

Topic which may not mach well to the scope of ICLR,  considereing that the authors' optimization is not about optimization on learning and/or representation but about a different type of optimization studied in the operations research/heuristic search communities

Poor presentation because of the lack of concrete examples that clearly illustrate the problems as well how their algorithm behaves with step-by-step traces e.g., a trace of priority queues with a small, concrete heuristic function, which indicates how search efforts can be reduced

Core idea which might have already been existing, especially the divide-and-conquer strategy of Hirschberg's DP algorithm and Korf's unidirectional and bidirectional A* in multiple sequence alginment

**Questions:**

How is the idea of compact A* similar to the existing approaches studied in multiple sequence alignment?

---

> ### Author Response · Authors · 2025-11-12
> **The sequence alignment problem finds a path of arbitrary length**
>
> Thank you for the thorough review.
>
> >the authors' optimization is about operations research/heuristic search and not about optimization on learning and/or representation.
>
> The sequence segmentation or histogram construction problem constructs an approximate representation of any signal or sequence.
>
> >Figure 1 does not not describe how each value in each square is calculated and it does not describe how calculations of the values are omitted by compact A*.
>
> Equation (1) describes exactly how each value in each square is calculated.
> Calculations are omitted by compact A* just like they are omitted by regular A*.
>
> >divide-and-conquer strategy is ... studied in solving the multiple sequence alignment problem.
>
> This divide-and-conquer strategy has been applied to the sequence alignment problem by Hirschberg and Korf, yet that problem finds an alignment path of arbitrary length. We apply the strategy to the harder problem of fixed-length path optimization. To appreciate the difference, we note that a sequence alignment problem corresponding to fixed-length path optimization would call to find an alignment between two input strings that chooses exactly a predefined number of $L$ characters from each of those strings.

---

### Official Review · Reviewer_jnNN · 2025-11-01

**Soundness:** 3
**Presentation:** 2
**Contribution:** 2
**Rating:** 4
**Confidence:** 3

**Summary:**

The paper considers the problem of lowest-cost path finding under fixed-length constraint, and proposes Compact-A*, a framework for space and time efficient optimization of fixed-length solutions. To control the size of the priority queue, the paper considers two containment mechanisms, using either depth-first search or a new earliest-first search. The paper applies the proposed framework to two problems, namely Viterbi decoding and V-optimal histogram computation. The experiments show significant improvement over dynamic programming baselines.

**Strengths:**

Strengths:
----
- New approach for fixed-length lowest-cost path search
- Theoretical guarantees are provided for the proposed algorithms
- Experiments show significant improvement over the dynamic programming baselines

**Weaknesses:**

Weaknesses:
----
- The extent and novelty of the technical contribution is not entirely clear to me: the base “Compact A*” algorithm seems like a limited-depth Dijkstra and not a new algorithm. It may be the first time such approach is applied to the two tasks considered but I am not convinced that this should be considered as a novel algorithm. I am also not sure why is it called “Compact A*” and not “Compact Dijkstra” as there is no heuristic function, only cost function (effectively a zero heuristic)?
- The algorithms for controlling the queue size (DFS/EFS) are not sufficiently described:
	* It seems that the DFS is not an enhancement of the current approach but the algorithm just switches to a DFS once it reaches the limit. Is that correct?
	* The impact on worst-case runtime complexity of the whole approach when using DFS/EFS strategies is not clear.
	* It is not clear stated whether the provided proofs of correctness take into account these strategies (the proofs do not seem to cover these settings).
- I found the separation between the base algorithm and the two instantiations of it (MINT/TECH) very confusing. In particular, the description of MINT is longer than the description of Compact A* with some repetition. Also not clear why propositions need to be proven for MINT rather than the base Compact A* framework or why the bidirectional variant is associated with MINT and not more broadly with the base algorithm.

**Questions:**

Please see weaknesses above for my questions and concerns

---

> ### Author Response · Authors · 2025-11-12
> **The problems we address are not addressed by limited-depth Dijkstra**
>
> Thank you for the thorough review.
>
> >the base “Compact A*” algorithm seems like a limited-depth Dijkstra
>
> Compact A* explores depth that has to match exactly a given fixed value, while limited-depth Dijkstra explores depth limited by a maximum value. This distinction is crucial, as a fixed depth value calls for tabulating solutions for each intermediate depth.
>
> >why is it called “Compact A*” and not “Compact Dijkstra” as there is no heuristic function
>
> A heuristic function exists in MINT Bound and TECH Bound algorithms.
>
> >the algorithm just switches to a DFS once it reaches the limit. Is that correct?
>
> The algorithm switches to DFS once it reaches a size limit and returns to regular BFS operation once the DFS iteration completes.
>
> >worst-case runtime complexity when using DFS/EFS strategies is not clear.
>
> The DFS strategy incurs $O(n^2 L^2)$ time in the worst case, as it may perform a DFS over O(nL) subsequent tokens for each of O(nL) tokens; in practice we do not encounter a large overhead. The EFS strategy, on the other hand, retains the standard $O(n^2 L)$ time complexity.
>
> >It is not clear stated whether the provided proofs of correctness take into account these strategies.
>
> The proofs hold under the memory-efficient strategies, as these strategies retain the calculation of the same exact objective.
>
> >why propositions need to be proven for MINT rather than the base Compact A* framework or why the bidirectional variant is associated with MINT and not more broadly with the base algorithm.
>
> The properties of algorithms and bidirectional variants need to be proven for each problem, as each specific problem requires attention to its own objective function and bounding methods.

---

### Note · Authors · 2026-01-19

I have read and agree with the venue's withdrawal policy on behalf of myself and my co-authors.